# Predictors of COVID-19 Vaccine Hesitancy: Socio-Demographics, Co-Morbidity, and Past Experience of Racial Discrimination

**DOI:** 10.3390/vaccines9070767

**Published:** 2021-07-09

**Authors:** Elena Savoia, Rachael Piltch-Loeb, Beth Goldberg, Cynthia Miller-Idriss, Brian Hughes, Alberto Montrond, Juliette Kayyem, Marcia A. Testa

**Affiliations:** 1Emergency Preparedness Research Evaluation & Practice (EPREP) Program, Division of Policy Translation & Leadership Development, Harvard T.H. Chan School of Public Health, 90 Smith Street, Boston, MA 02115, USA; piltch-loeb@hsph.harvard.edu (R.P.-L.); amontrond@hsph.harvard.edu (A.M.); testa@hsph.harvard.edu (M.A.T.); 2Department of Biostatistics, Harvard T.H. Chan School of Public Health, 677 Huntington Avenue, Boston, MA 02115, USA; 3Jigsaw, Google LLC, 82th 10th Ave, New York, NY 10024, USA; bethgoldberg@google.com; 4Center for University Excellence (CUE) and Polarization and Extremism Research and Innovation Lab (PERIL), American University, 4400 Massachusetts Avenue NW, Washington, DC 20016, USA; cynthia@american.edu (C.M.-I.); bhughes@american.edu (B.H.); 5Belfer Center for Science and International Affairs, Harvard Kennedy School, Harvard University, 79 John F. Kennedy Street, Cambridge, MA 02138, USA; Juliette_Kayyem@hks.harvard.edu; 6Massachusetts Association of Health Boards, 20 Walnut Street, Suite 110, Wellesley, MA 02481, USA

**Keywords:** COVID-19, vaccine hesitancy, discrimination

## Abstract

The goal of this study is to explore predictors of COVID-19 vaccine hesitancy, including socio-demographic factors, comorbidity, risk perception, and experience of discrimination, in a sample of the U.S. population. We used a cross-sectional online survey study design, implemented between 13–23 December 2020. The survey was limited to respondents residing in the USA, belonging to priority groups for vaccine distribution. Responses were received from 2650 individuals (response rate 84%) from all 50 states and Puerto Rico, American Samoa, and Guam. The five most represented states were California (13%), New York (10%), Texas (7%), Florida (6%), and Pennsylvania (4%). The majority of respondents were in the age category 25–44 years (66%), male (53%), and working in the healthcare sector (61%). Most were White and non-Hispanic (66%), followed by Black and non-Hispanic (14%) and Hispanic (8%) respondents. Experience with racial discrimination was a predictor of vaccine hesitancy. Those reporting racial discrimination had 21% increased odds of being at a higher level of hesitancy compared to those who did not report such experience (OR = 1.21, 95% C.I. 1.01–1.45). Communication and logistical aspects during the COVID-19 vaccination campaign need to be sensitive to individuals’ past-experience of racial discrimination in order to increase vaccine coverage.

## 1. Introduction

Within a year, the SARS-CoV-2 pandemic spread worldwide infecting millions of individuals and causing thousands of deaths. Under the federal Operation Warp Speed program, administered by the U.S. Department of Health and Human Services, $10 billion dollars were invested in six candidate vaccines [1]. In November 2020, Pfizer-BioNTech [2] and Moderna [3] reported that their much-anticipated vaccines demonstrated over 90% effectiveness in protecting people from the disease. Both vaccines were developed and tested at record speed and given U.S. Food and Drug Administration (FDA) emergency use authorization (EUA) in December 2020 [4,5]. Vaccine distribution of the Pfizer/BioNTech vaccine began on 8 December 2020. Two months prior to the approval of the vaccines, the Centers for Disease Control and Prevention (CDC) released its interim playbook for jurisdictional operations which outlined a phased approach to COVID-19 vaccination starting from those most at risk due to their job, age, and health status [6]. 

While the scientific efforts to produce the vaccine have been successful, the delivery of the vaccine has faced vast logistical, distribution, and communication challenges, the latter mostly related to hesitancy of individuals to take the vaccine. Immunization programs are only successful when there are high rates of acceptance and coverage. Addressing vaccine hesitancy, while delivering billions of doses across the world, is and will continue to be one of the greatest public health risk communication efforts ever undertaken. As such, it is critical to understand the reasons why specific segments of the population are more hesitant than others to accept the vaccine and address those reasons when implementing distribution and communication plans [7]. 

A recent review of 39 U.S. based polls shows that though there is potential for a majority of Americans to take COVID-19 vaccines, many people are still making up their minds [8]. This review concludes that while there is a good understanding that safety, effectiveness, and the desire of going back to a normal life are overall major motivating factors, communication approaches should be customized to the group that is undecided within a population. Polls conducted prior to the approval of the vaccines, as well as more recent surveys, indicate that in particular Black respondents are less likely to accept a potential COVID-19 vaccine [7,8,9,10]. This low acceptance is consistent with historical disparities in influenza immunization behavior and perceptions in the U.S. population, with Black adults significantly less likely to receive the influenza vaccine than White adults [11,12,13]. This is particularly concerning considering that Black individuals shoulder a disproportionate burden of many chronic conditions, placing them at a higher risk for complications from preventable diseases like influenza [14] and now for COVID-19. Research on the racial disparities in influenza immunization rates in the general population have identified several psychosocial and behavioral factors associated with vaccine uptake including: perceived risk, trust, vaccine attitudes, social norms, and experiences of racism [15,16]. The goal of this study is to explore the predictors of COVID-19 vaccine hesitancy including socio-demographic factors, co-morbidity, risk perception, and more specifically to investigate the role of past-experience with discrimination in predicting hesitancy, in particular among those identified as priority groups for vaccination. To our knowledge, this is the first time that experience with discrimination has been studied in relation to vaccine hesitancy. The study is based on a rapid survey conducted when the Pfizer-BioNTech and Moderna vaccines were approved and the ultimate goal is to inform public officials on how to enhance vaccine communication efforts during the vaccination campaign.

## 2. Materials and Methods

### 2.1. Patient and Public Involvement

There were no patients involved in the study.

### 2.2. Study Design

We used a cross-sectional online survey study design. The survey was implemented via mobile phones by the use of the survey platform Pollfish, and it was limited to respondents over 18 years of age residing in the USA. Similar to third-party advertising companies, Pollfish pays mobile application developers to display and promote the surveys to their users. To incentivize participation, relatively small monetary reimbursements are provided to randomly selected users who complete the surveys. An initial survey instrument draft was implemented for cognitive testing with 20 individuals, and the survey was subsequently revised after feedback to include 36 questions. Questions and response choices were kept short using “yes/no” or Likert-type and rating scales to facilitate completion by the use of mobile phones. The survey was launched on 13 December and closed on 23 December 2020. A screening question was used to identify respondents belonging to one of 19 job categories that were identified as priority groups for vaccine distribution based on national guidance available at the time of the survey [6]. The study protocol and survey instrument were approved by the Harvard T.H. Chan School of Public Health Institutional Review Board. A copy of the survey instrument can be found in Appendix A.

### 2.3. Dependent Variable

Multivariable ordinal regression was undertaken to model the underlying construct of vaccine hesitancy measured by the creation of a Likert-type scale. Respondents were asked how likely they would be to take a COVID-19 vaccine if offered to them at no cost within two months. Answer options were ordered as follows: very likely (1), somewhat likely (2), would consider it after two months (3), not sure (4), somewhat unlikely (5), very unlikely (6). Results were interpreted with a range of values from 1 (low hesitancy) to 6 (high hesitancy) maintaining the original order the answer options without grouping the answers into categories. 

### 2.4. Independent Variables

Independent predictor variables included socio-demographics such as age, gender, race, level of education, and employment status. Other predictors included job type (working in the healthcare sector vs. other priority groups for vaccination), having had a diagnosis of COVID-19 (with no symptoms, mild or severe symptoms), clinical risk of severe consequences from COVID-19, risk perception of contracting the disease or infecting others, and past experience with discrimination. Risk perception was measured by asking respondents to report their level of concern with contracting COVID-19 at work, outside their work environment, and infecting family members or friends. A factor analysis was performed to assess the structure of the risk perception questions, and as a result a scale was created with scores ranging from 0 to 6, with lower values indicating lower risk perception. Kaiser–Meyer–Olkin (KMO) measure of sampling adequacy was used to test for the suitability of the data for factor analysis and Cronbach’s alpha to assess the reliability of the scale. The respondents’ clinical risk for severe consequences from COVID-19 was measured by asking about the underlying health conditions most frequently associated with severe disease or death (diabetes, cardiovascular disease, obesity, pulmonary disease, immunocompromised status, rheumatological condition, or cancer), responses were converted into a dichotomous variable describing presence of at least one comorbidity vs. absence of comorbidities. Finally, respondents were asked about past experience with unfair treatment they attributed to their race, religion, gender or sexual orientation using an adaptation of the discrimination scale developed by Sternthal, M.J. et al. [16]. This scale includes six questions on unfair treatment experienced in the work environment, at school, by a police officer and by financial institutions (i.e., bank loan). The adaptation consisted of adding a question about unfair treatment by a physician or nurse and by limiting the cause of the unfair treatment to race, religion, gender, and sexual orientation. 

### 2.5. Statistical Analyses

We first performed descriptive statistics for each variable. We then applied simple and multiple ordinal regression models to study the association between the independent variables and COVID-19 vaccine hesitancy (dependent variable). We tested for bivariate associations between each predictor (age, gender, race, education, employment status, job type, having had a diagnosis of COVID-19, clinical risk profile, risk perception, and experience of discrimination) and the dependent variable, by means of ordinal and logistic regression using a *p*-value < 0.05 as cut-off for inclusion of the independent variables in the multiple regression model. We tested the parallel regression assumption by means of the *Brant* test for the ordinal logistic model which did not show statistical significance. The goodness-of-fit of the multiple variables model was tested by the use of the Hosmer-Lemeshow and Pulstenis-Robinson tests. The Stata Statistical Software 16 was used.

## 3. Results

### 3.1. Socio-Demographic Characteristics of the Study Population 

Responses were received from 2.650 respondents (response rate 84%) from all 50 states and the territories of Puerto Rico, American Samoa, and Guam. Descriptive statistics are given in Table 1. The five most represented states were California (13%), New York (10%), Texas (7%), Florida (6%), and Pennsylvania (4%). Sixty-six percent of respondents were age 25–44 years with median age 37 years, 53% were male, and 61% were working in the healthcare sector. The majority of respondents were white and non-Hispanic (66%) and others were Black non-Hispanic (14%) and Hispanic (8%). Respondents were highly educated with 31% having a graduate-level degree, and 86% were employed at the time of the survey.

### 3.2. Previous COVID-19 Diagnosis, Clinical Risk, and Risk Perceptions

As shown in Table 2, 24% of the sample respondents reported having had a prior diagnosis of COVID-19, 83% of whom had no or mild symptoms. Analysis of the clinical risk profile for severe consequences of COVID-19 indicated 26% of respondents reporting one of the seven conditions associated with greater risk, 5% reported two, and 2% reported three conditions or more. Fifty-eight percent of respondents were very concerned about getting infected at work, 48% were very concerned about contracting the disease outside the work environment, and 62% were very concerned about the possibility of infecting family members or friends. The factor analysis of these three risk perception questions resulted in one factor with eigenvalue >1, KMO = 0.7, alpha = 0.8. Based on the factor analysis results, a summative score was created to describe overall risk perception ranging from 0 (low risk) to 6 (high risk), and subsequently three categories of risk perception were created including: low risk perception (up to the 25th percentile), medium risk perception (25th < 75th percentile), and high-risk perception (≥75th percentile). Fifty-five percent of respondents were in the high-risk perception category, 31% in the medium risk category and 14.5% in the low risk.

### 3.3. Past Experience with Discrimination and Vaccine Hesitancy

As shown in Table 2, 63% of respondents reported having experienced at least once in their lifetime unfair treatment because of their race (34.5%), religion (12%), gender (21%), or sexual orientation (14%). Experience with unfair treatment due to race was reported by all race groups, 62% of Black respondents, 50% of those reporting two or more races, 49% of Hispanic, 45% of Asian, and 25% of white respondents. Experience with unfair treatment due to sexual orientation was reported by 50% of respondents who did not self-identify either as male or female, 23% identified as female, and 18% as male. Unfair treatment due to gender was reported by 40% of those self-identifying as neither male or female, 11% identifying as female, and 16% as male. In terms of type of experience Black and Hispanic individuals and those of two or more races reported the most discrimination as shown in Figure 1. Forty percent of the sample reported that they would be very likely to take the COVID-19 vaccine, if offered within two months from the time of the survey. In contrast, 13% said they were very unlikely to take it. The remaining 47% expressed various degrees of hesitancy with 15% responding that they would consider taking the vaccine in the future. 

### 3.4. Logistic Regression Models

Results of the simple and multivariable regressions are shown in Table 3. The goodness-of-fit tests resulted in Hosmer-Lemeshow (*p* = 0.74) and Pulstenis-Robinson (*p* = 0.33) tests. In the simple regression models (bivariate analysis) several variables were significantly associated with vaccine hesitancy. Female respondents had 25% decreased odds of reporting a higher level of hesitancy compared to male respondents (OR = 0.85, 95% C.I. 0.74–0.98). Respondents with some college education had 34% decreased odds of being at a higher level of hesitancy compared to individuals with less than a high school degree (OR = 0.66, 95% C.I. 0.44–0.99). Respondents reporting their race as Black and non-Hispanic had 1.22 times the odds of being at a higher level of hesitancy compared to any other race group (OR = 1.22, 95% C.I. 1.01–1.48). Those with a high-risk perception of contracting COVID-19 or of infecting a family member or friend had 1.30 times the odds of being at a higher level of hesitancy compared to those not having such concerns (OR = 1.30, 95% C.I. 1.06–1.60). Respondents who had COVID-19 with severe symptoms were more hesitant about taking the vaccine with 1.42 times the odds of being at a higher level of hesitancy compared to those who did not experience the disease at all (OR = 1.42, 95% C.I 1.01–1.99). Finally, those who experienced unfair treatment attributed to either their race, religion, gender, or sexual orientation had 1.19 the odds of being at a higher level of hesitancy compared to those who did experience discrimination due to the above-mentioned reasons (OR = 1.19, 95% C.I. 1.03–1.37). 

When the specific reasons for the perceived discrimination were analyzed as independent variables, racial discrimination was the only variable with a significant association with vaccine hesitancy. Those who experienced racial discrimination had 1.3 times the odds of being at a higher level of hesitancy compared to those who had not reported experiencing this type of discrimination (OR = 1.30, 95% C.I. 1.12–1.50) (see Figure 1). For the multivariable model, of vaccine hesitancy, the overall LR chi-square test statistics was significant (χ^2^, *p* < 0.01). Brant test *p*-value resulted 0.68. 

In the multivariable models, the only variable associated with vaccine hesitancy was experience of racial discrimination. Individuals with past experience had 21% increased odds of being at a higher level of vaccine hesitancy compared to those who did not report such experience. (OR = 1.21, 95% C.I. 1.01–1.45). The most frequently reported racial discrimination situation r was abuse from a police officer (15%), followed by having been denied a job or unfairly fired (13%), discouragement in pursuing an education was experienced by 11% of respondents. While all racial groups reported experience with unfair treatment due to their race, Black and Hispanic respondents and those of two or more races reported this experience most frequently.

## 4. Discussion

COVID-19 vaccine hesitancy has developed in a context where many showed fatigue due to the pandemic mitigation strategies, seeing them as ineffective, and in some cases even punitive. High acceptance of COVID-19 vaccines is critical to ending the pandemic especially among population groups for which high transmission rates have been recorded. Based on historical immunization data and recent polls, vaccine hesitancy is higher among Black persons compared to White persons [9,10,11,12,13,14,15,16,17]. The low likelihood of getting a COVID-19 vaccine among Black persons is especially concerning because of the high rates of transmission in Black communities [16]. Policymakers and public health professionals need to implement strategic plans to insure the vaccine reaches all Americans—particularly people of color belonging to the priority groups for the vaccination. Their concerns must be addressed throughout the course of the vaccination campaign. 

Public information and warning is one of the preparedness capabilities that public health agencies across the country will need to implement to support the vaccination campaign [18]. This capability entails the implementation of systems and procedures to mobilize communication activities such as fact gathering, rumor control, message testing, monitoring of media, social-media outlets and public opinions, and ultimately the publishing of content across print, Internet, social, and other media. They can also provide support to spokespersons by developing talking points, speeches, and visuals. 

The results from our study emphasize the need to potentiate public opinions’ monitoring strategies by gathering information on concerns and reasons for hesitancy towards the COVID-19 vaccine that get to the roots of such hesitancy, beyond the use of responses related to safety and effectiveness. The ultimate goal is to provide information that allows risk communicators and spokespersons to do a better job in targeting communication efforts to individuals’ informational needs, concerns, and past experiences. This study has the advantage of focusing on individuals belonging to priority groups receiving the vaccine. This is important as a successful vaccination campaign must demonstrate initial acceptance by the first to be vaccinated. Early adopters of immunization can have a strong influence on the likelihood that others will accept the vaccine and will be compliant with the immunization recommendations. In particular, our sample included a large fraction of individuals working in the healthcare sector who could play a key role in advocating for the vaccine among the general population. 

To our knowledge, our study is the first to date to include a measure of past experience of discrimination as a predictor of COVID-19 vaccine hesitancy, enriching current knowledge on the relationship between vaccine hesitancy and race. This result is important to inform communication and logistical aspects of the COVID-19 vaccination campaign. While our study does not specifically address the association between experience of discrimination and trust in the organizations in charge of the vaccination efforts, previous research demonstrates that trust remains distinct from vaccine confidence in both the general and flu vaccine. Specific models show that trust in information sources alone does not explain the observed relationship between race and vaccination beliefs [19,20,21] and that Black persons have a different type of mistrust related to the COVID-19 vaccine compared to White persons: a mistrust in the government entities’ motives rather than in their competence [22]. A discussion regarding the importance of developing strategies so that members of the Black and other communities not only trust that the COVID-19 vaccines are safe and effective, but also believe that the organizations offering them are trustworthy was initiated back in 2020 when low rates of the vaccine trial participation in subgroups, including Black communities, were reported [22]. While we recognize that trust is a complex construct which we have not investigated in our study, we also believe that our results indicate that future research should focus in understanding how experience of discrimination might be a mediator of mistrust in the vaccine and the system delivering it, as suggested by other authors [19,20]. 

It is certainly clear that during the short timeframe of this vaccination campaign, even with the best of intentions, policy makers and public health practitioners will not be able to undo centuries of distrust based on unfair treatment and discrimination experienced by specific segments of the population in sectors such as health care, education, finance, and safety. However, they can be sensitive to individuals’ concerns and past experiences and educate clinicians and spokespersons on historical facts, avoiding use of law enforcement to surveil the safety of vaccination sites, engaging individuals from Black communities in vaccination efforts, and educating policy makers and vaccine distribution planners on the potential root causes of mistrust. We believe that enhancing the vaccine uptake among Black Americans requires much more than disseminating facts about safety—it is likely to require overcoming barriers of mistrust in the system. Policy makers and public officials need to start by acknowledging, appreciating, and discussing public concerns. Labeling those hesitant about the vaccine as conspiracy theorists or individuals unwilling to prevent the spread of the disease, may be counterproductive when hesitancy is rooted in a history of unfair treatment which will not be overturned by denying the existence of fear and doubts. Given the complexity of vaccine hesitancy in general and the limited evidence available on how it can be addressed, communication and distribution strategies should be carefully tailored according to the target population, their reasons for hesitancy, and the specific context as demonstrated by a recent literature review [21]. The tailoring of messages is important to address communication inequalities in individual or group specific exposure and reactions to public health communication messages [22,23,24,25], which may lead to further enhance existing disparities across segments of the population in the ability to comply with recommended preventive behaviors. 

Opinion surveys, at the time of crisis, are a tool to understand people’s concerns so that such concerns can be addressed and taken into consideration when developing better communication efforts. These concerns are currently further fueled by an emotional dimension driven by social isolation, daily life restrictions, and difficulties experienced during the pandemic. Public health agencies need to enhance their public information capabilities to address multiple dimensions of the vaccine communication strategy in order to be successful and meet growing needs of information and reassurance across a diverse audience.

### Study Limitations

Because we used a cross sectional study design, the timing of the survey must be considered in interpreting the results. The survey was fielded in December 2020 when vaccines were announced but not yet available to the public. Due to the evolving epidemiology of the disease and the developing public communication and vaccine distribution efforts, the predictors of vaccine hesitancy are likely to change over time. In particular, we acknowledge that in our study we did not find a statistical significant association between vaccine hesitancy and risk perception of contracting COVID-19 which is likely to be relevant for future studies. From a methodological point of view the dependent variable “vaccine hesitancy” can be categorized in different ways. For the purpose of this study we elected to capture different degrees of hesitancy and use ordinal logistic regression to do so. We believe this is a strength rather than a weakness but we do acknowledge that different categorizations may lead to different results and that ROC analysis should be applied to study potential for misclassification [26]. We also remind the reader that our sample is not a representative sample of the US population as such study results are not generalizable outside the study population. We purposefully focused on the demographic characteristics of the population eligible for the vaccine at the time of the survey because we believe vaccine hesitancy needs to be measured taking into consideration for whom and when the vaccine is available. While our sample included a distribution of racial-ethnic groups that allowed us to analyze predictors of vaccine hesitancy based on race it did not include a sufficient number of individuals over 65 which based on previous studies are more likely than others to accept the COVID-19 vaccine due the increased risk of severity in the elderly. Yet, the reason for such limitation is related to the fact that we were interested in surveying individuals eligible for the vaccine at the time of the survey, such eligibility was mainly based on job categories rather than age. Given changes in the vaccine distribution criteria, future studies should include a wider distribution of demographic categories. 

## 5. Conclusions

Results from this survey of a convenience sample of the US population show that past experience with discrimination is a predictor of vaccine hesitancy. This result is important to inform communication and logistical aspects during the COVID-19 vaccination campaign which need to be sensitive to individuals’ past experience with systemic unfair treatment by different types of institutions including law enforcement, education and healthcare.

## Figures and Tables

**Figure 1 vaccines-09-00767-f001:**
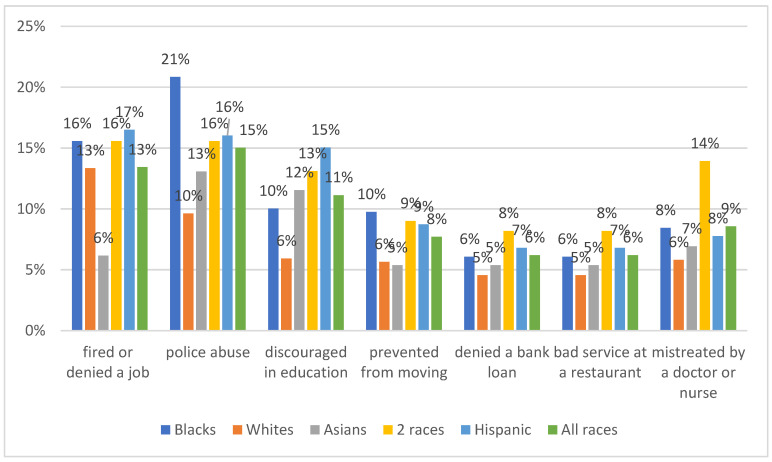
Experience of racial discrimination.

**Table 1 vaccines-09-00767-t001:** Socio-demographic characteristics of the study population.

Age	N (%)
18–24	354 (13.4%)
25–34	841 (31.7%)
35–44	906 (34.2%)
45–54	339 (12.8%)
55–64	152 (5.7%)
65–74	47 (1.8%)
≥75	11 (0.4%)
**Gender**	**N (%)**
Male	1417 (53.5%)
Female	1213 (45.8%)
Other	20 (0.7%)
**Education**	**N (%)**
Less than high school	92 (3.2%)
High school/GED	539 (20.3%)
Some college	579 (21.9%)
Bachelor’s degree	615 (23.3%)
Post-graduate degree	825 (31.3%)
**Race**	**N (%)**
White, non-Hispanic	1754 (66.2%)
Black, non-Hispanic	379 (14.3%)
Hispanic	206 (7.8%)
Asian, non-Hispanic	130 (4.9%)
2+ races	122 (4.6%)
Prefer not to say	40 (1.5%)
Other	19 (0.7%)
**Employment status**	**N (%)**
Paid employee	2032 (76.7%)
Self-employed	243 (9.2%)
On unemployment	101 (3.8%)
Not working-searching for work	96 (3.6%)
On paid leave or furloughed	41 (1.6%)
Retired	41 (1.6%)
Not working-and not looking for a job	39 (1.4%)
On disability or worker’s compensation	35 (1.3%)
Other	22 (0.8%)
**Job category (multiple choice question)**	**N (%)**
Hospital and emergency department workers	624 (23.5%)
Nursing home, long-term care, and home health care workers	413 (15.6%)
Public health workers	284 (10.7%)
Grocery store workers	283 (10.7%)
Teachers and school staff	251 (9.5%)
Food processing workers	222 (8.4%)
Emergency Medical Services workers	186 (7.0%)
Other health care workers	170 (6.4%)
Volunteer (i.e., CERT, MRC, Red Cross)	168 (6.3%)
Private transportation workers	156 (5.9%)
Sanitation workers	131 (4.9%)
Vaccine manufacturing workers	121 (4.6%)
Postal and shipping workers	120 (4.5%)
Pharmacy workers	117 (4.4%)
Correctional facilities workers	116 (4.4%)
Police or firefighters	116 (4.4%)
Vaccine distribution workers	95 (3.6%)
Other first responders	93 (3.5%)
Public transportation workers	90 (3.4%)

**Table 2 vaccines-09-00767-t002:** Comorbidity, risk perception, experience of discrimination, and vaccine hesitancy of the study population.

Co-Morbidity (Diabetes, Obesity, Rheumatological Disease, Immunocompromised Status, Cancer, Cardiovascular Disease, Chronic Respiratory Disease)	N (%)
No medical condition	1764 (66.6%)
One medical condition	685 (25.8%)
Two or more medical conditions	201 (7.6%)
**Have you been diagnosed with COVID-19?**	**N (%)**
No	1961 (74%)
I am not sure	57 (2.2%)
Yes, with no symptoms	266 (10%)
Yes, with mild symptoms	259 (9.8%)
Yes, with severe symptoms	107 (4%)
**Experience of unfair treatment**	**N (%)**
Attributed to any of the following reasons: race, religion, gender and sexual orientation	1680 (63.4%)
Race was the only reason or one of the reasons	915 (34.5%)
Religion was the only reason or one of the reasons	318 (12%)
Gender was the only reason or one of the reasons	549 (20.7%)
Sexual orientation was the only reason or one of the reasons	361 (13.6%)
**How concerned are you about any of the following situations?**	**N (%)**
Contracting COVID-19 at work? (For example: hospital, office, and other work settings that are not your home)	
Very concerned	1542 (58.3%)
Somewhat concerned	792 (29.9%)
Not concerned	312 (11.8%)
Contracting COVID-19 outside of work? (For example: at the grocery store, when you are using transportation, or in other aspects of your daily life)	
Very concerned	1266 (47.9%)
Somewhat concerned	1007 (38.1%)
Not concerned	371 (14%)
Infecting your family or friends with COVID-19?	
Very concerned	1653 (62.5%)
Somewhat concerned	664 (25.1%)
Not concerned	326 (12.4%)
**COVID-19 overall risk perception**	**N (%)**
Low risk	382 (14.5%)
Medium risk	813 (30.1%)
High risk	1439 (54.6%)
**If you were offered a COVID-19 vaccine within two months from now-at no cost to you- how likely are you to take it?**	**N (%)**
Very likely (low hesitancy)	1059 (40%)
Somewhat likely	523 (19.7%)
I would not take it within 2 months but would consider it later on	188 (7.1%)
Not sure	388 (14.6%)
Somewhat unlikely	153 (5.8%)
Very unlikely (high hesitancy)	339 (12.8%)

**Table 3 vaccines-09-00767-t003:** Association between independent variables and vaccine hesitancy: simple models and multiple variable models.

	Simple Models		Multiple Model	
Independent Variable	OR	95% C.I.	OR	95% C.I.
**Age**				
18–24	**-**	**-**	**-**	**-**
25–34	1.07	0.86–1.34	-	-
35–44	0.99	0.80–1.24	-	-
45–54	1.03	0.79–1.35	-	-
55–64	0.97	0.69–1.36	-	-
65–74	0.75	0.43–1.32	-	-
≥75	1.16	0.41–3.24	-	-
**Gender**				
Female vs. male	0.85 *	0.74–0.98	0.91	0.78–1.05
Other than female or male vs. male	0.59	0.26–1.33	0.62	0.27–1.42
**Employment status**				
Paid employee and self-employed vs. other categories	1.14	0.94–1.39	-	-
**Education**				
Less than high school	**-**	**-**	**-**	**-**
High school/GED	0.78	0.52–1.15	0.78	0.52–1.17
Some college	0.66 *	0.44–0.99	0.68	0.45–1.01
Bachelor’s degree	0.77	0.52–1.13	0.78	0.52–1.16
Post-graduate degree	0.97	0.66–1.43	0.95	0.64–1.40
**Race**				
White non-Hispanic vs. all other races	0.94	0.81–1.1	-	-
Black non-Hispanic vs. all other races	1.22 *	1.00–1.48	1.18	0.96–1.44
Asian non-Hispanic vs. all other races	0.87	0.63–1.20	-	-
Hispanic vs. all other races	0.92	0.71–1.20	-	-
**Type of job**				
Healthcare sector employee vs. other job categories	1.09	0.94–1.25	-	-
**Medical conditions**				
No medical condition	-	-	-	-
One medical condition	1.06	0.90–1.24	-	-
Two medical conditions	1.23	0.90–1.69	-	-
Three or more medical conditions	0.75	0.46–1.22		
**Risk perception**				
Low risk perception	-	-	-	-
Medium risk perception	1.14	0.92–1.42	1.10	0.88–1.38
High risk perception	1.30 *	1.06–1.60	1.18	0.95–1.47
**COVID-19 diagnosis**				
No diagnosis	-	-	-	-
Not sure	1.05	0.65–1.68	1.01	0.62–1.63
Yes–no symptoms	1.02	0.81–1.29	0.89	0.69–1.13
Yes–mild symptoms	1.13	0.89–1.43	1.02	0.80–1.29
Yes–severe symptoms	1.42 *	1.01–1.99	1.27	0.90–1.79
**Experience of unfair treatment**				
Attributed to any of the following reasons: race, religion, gender or sexual orientation	1.19 *	1.03–1.37	0.97	0.82–1.16
Race was the only reason or one of the reasons	1.30 **	1.12–1.50	1.21 *	1.01–1.45
Religion was the only reason or one of the reasons	1.21	0.98–1.49	-	-
Gender was the only reason or one of the reasons	0.97	0.83–1.14	-	-
Sexual orientation was the only reason or one of the reasons	0.97	0.80–1.19	-	-

* *p* < 0.05. ** *p* < 0.001.

## Data Availability

Data are available at: https://github.com/esavoia123/Vaccine-hesitancy-data-Dec-2020-USA.git (accessed on 6 July 2021).

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
