# Peer review of "Predictors of COVID-19 Vaccine Hesitancy: Socio-Demographics, Co-Morbidity, and Past Experience of Racial Discrimination"

_vaccines, 2021, doi:10.3390/vaccines9070767_

Round 1

Reviewer 1 Report

The paper concerns an important and relevant topic, and is well written.

It should be improved in two main ways:

  1. In terms of survey design, trying to have a sample of respondents that is more representative of the different American demographic groups.

2. From a methodological viewpoint, it is not clear how the response variable (hesitancy) has been binarised before running logistic regression. For robustness, the authors should compare alternative binarisations. This may improve the results of multiple regressions which are the moment are limited to racial groups. 

3. It is necessary to verify the goodness of fit of the model also in terms of predictive accuracy, dividing the sample in two, according to the cross validation paradigm.

4. More literature on covid19 contagion and on the impacts of vaccines and vaccine hesitancy on it should be included

Author Response

We appreciate the time dedicated by the reviewer in evaluating the statistical approach and overall presentation of the study. We believe all points suggested by the reviewer have been addressed and allowed us to improve the manuscript. 

Reviewer: In terms of survey design, trying to have a sample of respondents that is more representative of the different American demographic groups.

Authors: We agree with the reviewer that the sample is not representative of the variety of demographic groups in the USA. However, please note that the reason for such limitation is related to the fact that our interest was to survey individuals included in the priority groups for vaccine distribution at the time of the survey which was December 2020 (see methods).  As such most of these individuals – eligible for the vaccine- did not belong to all demographic groups in the USA.  We believe vaccine hesitancy is to be measured and interpreted at a given point in time based on  the availability of the vaccine and distribution strategy.  To address the reviewer's comment and make sure the reader is not confused about the characteristics of the sample we now have extensively emphasized and discussed this in the "limitation" section. 

Reviewer: From a methodological viewpoint, it is not clear how the response variable (hesitancy) has been binarised before running logistic regression. For robustness, the authors should compare alternative binarisations. This may improve the results of multiple regressions which are the moment are limited to racial groups. 

Authors: The variable is described in section 2.3 under methods. We now explain that we did not collapse/group the answers which we agree with the reviewer was not clear before. As reported by SteeFisher et al. in the NEJM “Polls suggest that much of the U.S. public is currently undecided about whether to take a Covid-19 vaccine. This point is often overlooked, since interpreters of several recent polls have predicted that a majority will get vaccinated.  But these interpretations are missing an important detail: reporting of poll results often involves collapsing various categories of responses.” For this reason and limitation of previous studies we decided not to collapse the categories of hesitancy, to be able to capture various degrees of hesitancy. Please note that we used ordinal logistic regression.   In regards to the limitation of using the dependent variables based on one type of measurement approach. We have now clarified in the introduction that among the socio-demographic variables investigated, past experience with discrimination and its association with vaccine hesitancy was of high interest for us and to our knowledge an association never investigated before in relation to COVID-19 vaccine hesitancy. Most studies have focused on identifying racial and ethnic differences in hesitancy without introducing this variable. We believe this is the innovative aspect of our approach and the current dependent variable and model supported out approach and preliminary hypothesis.

Reviewer: It is necessary to verify the goodness of fit of the model also in terms of predictive accuracy, dividing the sample in two, according to the cross validation paradigm.

Authors: We now report the Goodness of Fit of the model. Goodness of fit was tested with Hosmer-Lemeshow and PR = Pulkstenis-Robinson tests leading to p=0.74 and p=0.33 respectively. 

Reviewer: More literature on covid19 contagion and on the impacts of vaccines and vaccine hesitancy on it should be included

Authors: We agree with the reviewer that the literature cited in the introduction is not up to date, this is mainly due to the fact that the manuscript was written a couple of months ago and a lot has been published since then. We have now extensively updated and revised the introduction and in particular we have included a review of 39 polls which is a good summary of what we know up to date on vaccine hesitancy. 

Reviewer 2 Report

Owing to ending the COVID-19 pandemic, it is important to know predictors of COVID-19 vaccine hesitancy including socio-demographic factors, co-morbidity, risk perception and past-experience with discrimination, in particular among those identified as priority groups. In line with this points of view, authors are well delineate how to enhance vaccine communication efforts via the vaccination campaign. There are some minor point. Please check the typo and spacing.

Author Response

We appreciate the reviewer's comments. We have now extensively edited the manuscript for grammar, typos and space. Thank you. 

Round 2

Reviewer 1 Report

The authors have satisfactorily addressed many of my previous comments. It remains one, about predictive accuracy. Most predictive accuracy tools rare based on the Receiver Operating Characteristic (ROC),  and the area under it (AUROC). The curve was recently found useful also to explain the relative contribution of each variable. The authors should include at least references, to these extensions,  for example:

and 

https://www.sciencedirect.com/science/article/pii/S0957417420308575

Author Response

We have included one of the two references as suggested by the reviewer and pointed to this limitation in our analysis. We believe the reviewer is raising an important issue related to misclassification which however should be object of a separate paper focused on methods related to how vaccine hesitancy is measured probably directed to a different audience. With this paper we are currently aiming to reach public health practitioners to help them interpret differences in vaccine acceptance across racial-ethnic groups.